# DIME: An Information-Theoretic Difficulty Measure for AI Datasets

**Peiliang Zhang**[*][†]
University of Pittsburgh
pez35@pitt.edu

**Huan Wang**[†]
Salesforce Research
huan.wang@salesforce.com

**Nikhil Naik**
Salesforce Research
nnaik@salesforce.com

**Caiming Xiong**
Salesforce Research
cxiong@salesforce.com

**Richard Socher**
Salesforce Research
rsocher@salesforce.com

## Abstract

Evaluating the relative difficulty of widely-used benchmark datasets across time and across data modalities is important for accurately measuring progress in machine learning. To help tackle this problem, we propose **DIME**, an information-theoretic **DI**fficulty **ME**asure for datasets, based on Fano's inequality and a neural network estimation of the conditional entropy of the sample-label distribution. DIME can be decomposed into components attributable to the data distribution and the number of samples. DIME can also compute per-class difficulty scores. Through extensive experiments on both vision and language datasets, we show that DIME is well-aligned with empirically observed performance of state-of-the-art machine learning models. We hope that DIME can aid future dataset design and model-training strategies.

## 1 Introduction

Empirical machine learning research relies heavily on comparing performance of algorithms on a few standard benchmark datasets. Moreover, researchers frequently introduce new datasets that they believe to be more challenging than existing benchmarks. However, we lack objective measures of dataset difficulty that are independent of the choices made about algorithm- and model-design. Moreover, it is also hard to compare algorithmic progress across data modalities, such as language and vision. So, for instance, it is difficult to compare the relative progress made on a sentiment analysis benchmark such as the Stanford Sentiment Treebank (SST) (Socher et al., 2013) and an image classification benchmark, like CIFAR-10 (Krizhevsky, 2009). With these challenges in mind, we propose a model-agnostic and modality-agnostic measure for comparing how difficult it is to perform supervised learning on a given dataset.

For i.i.d. dataset examples, we argue that the difficulty of a dataset can be decoupled into two relatively independent sources:(a) approximation complexity, the number of samples required to approximate the true distribution up to certain accuracy, and (b) distributional complexity, the intrinsic difficulty involved in modeling the statistical relationship between the labels and features.

We focus our analysis on the second source of the intrinsic difficulty in supervised learning, and consider the most common setting where features $X$ are continuous and the labels $Y$ are discrete. To provide a model-agnostic measure, we turn to the information-theoretic approach. Indeed, Fano's inequality suggests the lowest possible probability of the 0-1 error $\mathcal{P}_e$ is lower bounded by terms

---

[*]Work partially done while the author was a research intern at Salesforce Research.
[†]Equal Contribution.

34th Conference on Neural Information Processing Systems (NeurIPS 2020), Vancouver, Canada.

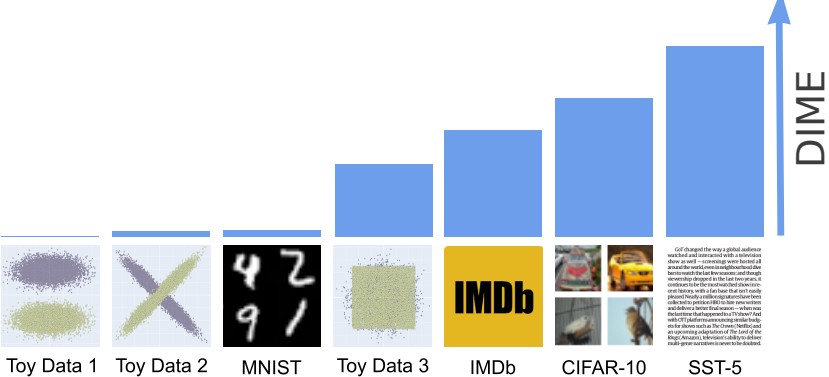

| Toy Data 1 | Toy Data 2 | MNIST | Toy Data 3 | IMDb | CIFAR-10 | SST-5 |

Figure 1: We propose DIME, a model- and modality-agnostic difficulty metric for datasets

related to the conditional entropy $\mathbb{H}(Y|X)$. Naturally we consider designing an estimator for this lower bound of $\mathcal{P}_e$ as a difficulty measure for learning $Y$ given $X$. To estimate $\mathbb{H}(Y|X)$, notice that $\mathbb{H}(Y|X) = \mathbb{H}(Y) - \mathbb{I}(X, Y)$, where $\mathbb{I}(X, Y)$ is the mutual information. Direct estimation of $\mathbb{I}(X, Y)$ is hard for the hybrid case where $X$ is continuous and $Y$ is discrete. Our key observation is that we can express $\mathbb{I}(X, Y)$ as a weighted average of the KL divergence between $X|Y = y$ and $X$ for each label $y$ (see equation (4)). Then we may use neural networks to approximate the KL divergence based on Donsker-Varadhan representation.

**Related Work:** Although conditional entropy and mutual information estimation have been extensively studied, research has focused on purely discrete or continuous data. Nair et al. (2006) were among the first to study the theoretical properties for the case of mixture of discrete and continuous variables. Ross (2014), Gao et al. (2017) and Beknazaryan et al. (2019) proposed approaches for estimating mutual information for the mixture case based on density ratio estimation (e.g.,binning, kNN or kernel methods), which is unsatisfactory for high dimensional data such as image and text. We use neural network estimation, inspired by Belghazi et al. (2018), to avoid these issues. However, we emphasize that the technique for estimating the mutual information in Belghazi et al. (2018) cannot be applied directly to the hybrid case, as argued before. More importantly, we are the first to connect the hybrid conditional entropy with the difficulty measure for datasets.

## 2 Designing a Dataset Difficulty Measure

### 2.1 Main Idea

For the continuous feature and discrete label setting, we use $(X, Y)$ to represent feature-label pairs and assume $X \in \mathcal{X} \subset \mathbb{R}^{d_x}$, and $Y \in \mathcal{Y} \subset \mathbb{Z}^+$. We denote the joint distribution of $(X, Y)$ as $\mathcal{P}_{\mathcal{X}\mathcal{Y}}$ and the marginal distributions as $\mathcal{P}_{\mathcal{X}}$ and $\mathcal{P}_{\mathcal{Y}}$ respectively. Throughout this paper, we always assume that the feature-label pairs in the datasets, both training and testing, are sampled i.i.d. from a static distribution $\mathcal{P}_{\mathcal{X}\mathcal{Y}}$. As $\mathcal{Y}$ is discrete, a natural measure that characterizes the difficulty of the data distribution is the best probability of the 0-1 error that can be achieved by any estimator.

**Definition 1.** *Model-Agnostic Error*

$$\mathcal{P}_e = \inf_f \mathcal{P}_{X,Y \sim \mathcal{P}_{\mathcal{X}\mathcal{Y}}} [f(X) \neq Y]. \tag{1}$$

While the computation of general $\mathcal{P}_e$ is almost impossible, Fano's inequality establishes a lower bound of $\mathcal{P}_e$ by terms related to the conditional entropy.

**Definition 2.** *Hybrid conditional entropy*

$$\mathbb{H}(Y|X) := \mathbb{E}_X \big[ - \sum_{y=1}^{|\mathcal{Y}|} \mathbb{P}(Y = y|X) \log \mathbb{P}(Y = y|X) \big], \tag{2}$$

*where* $\mathbb{P}(Y = y|X) := \mathbb{E}[\mathbf{1}(Y = y)|X]$.

**Theorem 1.** *Fano's inequality for continuous feature-discrete label case: Let $\mathcal{P}_e$ be the minimum error probability as defined in* (1). *Then we have*

$$\mathbb{H}(\mathcal{P}_e) + \mathcal{P}_e \log(|\mathcal{Y}| - 1) \geq \mathbb{H}(Y|X), \tag{3}$$

*where* $\mathbb{H}(\mathcal{P}_e) := -\mathcal{P}_e \log \mathcal{P}_e - (1 - \mathcal{P}_e) \log(1 - \mathcal{P}_e)$.

A detailed proof is given in Appendix C. Our proof is based on the proof of Lemma 2.10, Tsybakov (2004). Note that most of the proofs of Fano's inequality are designed for discrete random variables (e.g. Cover & Thomas (2012)).

## 2.2 Estimating the Lower Bound of $\mathcal{P}_e$

Motivated by Fano's inequality, we just need to estimate $\mathbb{H}(Y|X)$. We rewrite it as (see Appendix B for a proof)

$$\mathbb{H}(Y|X) = \mathbb{H}(Y) - \mathbb{I}(X,Y), \quad \text{with}$$

$$\mathbb{H}(Y) = -\sum_{y=1}^{|\mathcal{Y}|} \mathbb{P}(Y=y) \log \mathbb{P}(Y=y), \quad \mathbb{I}(X,Y) = \sum_{y=1}^{|\mathcal{Y}|} P(Y=y) \mathbb{KL}(X|Y=y||X). \tag{4}$$

**Remark 1.** *In some benchmark datasets with balanced classes (e.g., CIFAR-10 and MNIST), $\mathbb{H}(Y|X) = \log|\mathcal{Y}| - \frac{1}{|\mathcal{Y}|}\sum_{y=1}^{|\mathcal{Y}|} \mathbb{KL}(X|Y=y||X)$. This indicates that if a dataset has more classes and the distributions of the features for different classes are closer to each other on average, then $\mathbb{H}(Y|X)$ would be larger and thus pushes $\mathcal{P}_e$ larger, implying that the classification task on it would be harder due to the intrinsic distributional complexity.*

We now discuss the practical design of an estimator for $\mathbb{KL}(X|Y=y||X)$. (Notice that both of the random varibles are continuous.) Many non-parametric KL divergence estimators such as $k$NN based techniques (Noshad et al., 2017; Perez-Cruz, 2008; Wang et al., 2009) are not satisfying as they often rely on some knowledge of the internal dimension of the data manifold. As an alternative, we design a neural network-based estimator for KL divergence estimation, inspired by Belghazi et al. (2018). The estimator is based on the Donsker-Varadhan representation:

$$\mathbb{KL}(\mathcal{P}||\mathcal{Q}) = \sup_{T:\mathcal{X}\to\mathbb{R}} \mathbb{E}_{\mathcal{P}}[T] - \log \mathbb{E}_{\mathcal{Q}}[e^T], \tag{5}$$

for two probability measures $\mathcal{P}, \mathcal{Q}$. If we parameterize the function $T$ using a neural network $T_\theta$, we can obtain an estimator of $\widehat{\mathbb{KL}}(X|y||X)$ for each $y \in \mathcal{Y}$. Then by equation (4), we get $\widehat{\mathbb{H}}(Y|X)$, which further leads to $\widehat{\mathcal{P}_e}$ by (1), an estimator of the lower bound of $\mathcal{P}_e$. We report $\widehat{\mathcal{P}_e}$ as our datatset difficulty measure DIME.

In our empirical evaluation, we optimize the empirical average instead of the expectation. However this may cause issues such as overfitting. To mitigate this issue, we split the data into separate training and validation sets. The neural network models are trained on the training set and the estimation is made using the validation set. See Algorithm 1 in Appendix A.1 for details.

**Remark 2.** *Given that we are using a neural network model to estimate the KL divergence between $\mathcal{P}(X|y)$ and $\mathcal{P}(X)$, according to Equation (5), $\widehat{\mathbb{KL}}(X|y||X) \leq \mathbb{KL}(X|y||X)$. As a consequence, the estimated mutual information $\widehat{\mathbb{I}}(X;Y) \leq \mathbb{I}(X;Y)$, which leads to a larger $\widehat{\mathbb{H}}(Y|X)$. In the end, DIME could be larger as compared to true lower bound on $\mathcal{P}_e$. The hope is, if the function class of the neural network is large enough, the gap between $\widehat{\mathbb{KL}}(X|y||X)$ and $\mathbb{KL}(X|y||X)$ is small enough so that DIME won't deviate too much from the true lower bound of $\mathcal{P}_e$.*

# 3 Experiments and Results

## 3.1 Sanity-check Experiments

**Experiments with toy datasets:** We generate three toy datasets and report their DIME socres, as shown in Figure 1. The details are given in Appendix A.2, which contains a label corruption test as well.

## 3.2 Evaluating benchmark datasets in vision and language

We evaluate DIME using popular image and language domain datasets for various classification tasks. We use a simple multi-layer perceptron (MLP) with residual layers with ReLU activation functions, containing three hidden layers of 4096 neurons. (The details about the datasets and neural network we

| Corpus | #classes | $\widehat{\mathbb{H}}(Y)$ | $\widehat{\mathbb{I}}(X,Y)$ | $\widehat{\mathbb{H}}(Y|X)$ | DIME | SOTA Error |
|---|---|---|---|---|---|---|
| **Image Classification** | | | | | | |
| EMNIST (digits) | 10 | 2.303 | 2.255 | 0.048 | 0.006 | 0.002 |
| MNIST | 10 | 2.301 | 2.192 | 0.109 | 0.015 | 0.002 |
| EMNIST (letters) | 26 | 3.258 | 2.872 | 0.386 | 0.054 | 0.056 |
| EMNIST (bymerge) | 47 | 3.554 | 3.098 | 0.456 | 0.060 | 0.190 |
| Fashion-MNIST | 10 | 2.303 | 1.912 | 0.391 | 0.066 | 0.033 |
| EMNIST (byclass) | 62 | 3.679 | 3.126 | 0.553 | 0.072 | 0.240 |
| EMNIST (balanced) | 47 | 3.850 | 3.212 | 0.638 | 0.089 | 0.095 |
| SVHN | 10 | 2.223 | 1.436 | 0.786 | 0.159 | 0.010 |
| CIFAR-10 | 10 | 2.303 | 0.915 | 1.388 | 0.340 | 0.010 |
| CIFAR-100-Subclass | 10 | 2.303 | 0.503 | 1.800 | 0.504 | N/A |
| CIFAR-100 | 100 | 4.605 | 1.239 | 3.366 | 0.585 | 0.087 |
| Tiny ImageNet | 200 | 5.298 | 0.692 | 4.606 | 0.768 | 0.268 |
| FakeData | 10 | 2.303 | -0.003 | 2.306 | 0.900 | 0.900 |
| **Sentiment Analysis** | | | | | | |
| IMDb | 2 | 0.693 | 0.224 | 0.469 | 0.178 | 0.038 |
| SST-2 | 2 | 0.693 | 0.224 | 0.469 | 0.179 | 0.032 |
| SST-5 | 5 | 1.573 | 0.230 | 1.342 | 0.470 | 0.356 |
| **Text Classification** | | | | | | |
| DBPedia | 14 | 2.639 | 2.387 | 0.252 | 0.037 | 0.013 |
| TREC | 6 | 1.638 | 1.185 | 0.453 | 0.092 | 0.019 |
| YelpReview (Polarity) | 2 | 0.693 | 0.371 | 0.322 | 0.098 | 0.044 |
| AG News | 4 | 1.386 | 0.808 | 0.579 | 0.147 | 0.076 |
| **Language Modeling** | | | | | | |
| Penn Treebank | 42 | 2.966 | 1.876 | 1.090 | 0.165 | 1.083 (ppl) |

Table 1: We evaluate DIME on vision and language datasets and rank them by relative difficulty. Comparisons with prediction performance of state-of-the-art neural network models shows that DIME is roughly aligned with empirically observed performance. (ppl: perplexity) The Pearson and Kendall's Tau correlation coefficient between DIME and SOTA error are 0.7437 and 0.3470, respectively.

choose are given in Appendix A.3.) In addition to dataset statistics, we report the estimated entropy $\widehat{\mathbb{H}}(Y)$, $\widehat{\mathbb{I}}(X,Y)$, $\widehat{\mathbb{H}}(Y|X)$, and DIME. We also report the state-of-the-art error on the dataset from recent literature (Wan et al., 2013; Huang et al., 2018; Cubuk et al., 2019; Yang et al., 2019; Patro et al., 2018; Cer et al., 2018; Melis et al., 2019). From the results summarized in Table 1, we make the following empirical observations:

First, DIME aligns well with the state-of-the-art results on the benchmark datasets, indicating that the relative difficulty of datasets is reflected in the performance of latest machine learning algorithms. However, notably, DIME is lower than the reported state-of-the-art error for all the variants of EMNIST. This could indicate that since EMNIST is a relatively new, less-investigated dataset, we might see model performance improving over time.

Second, a larger number of classes lead to a higher value of DIME in general, which is expected given its dependence on $\widehat{\mathbb{H}}(Y)$. But $\widehat{\mathbb{H}}(Y)$ does not always dominate the value of the measure. For example, MNIST has 10 classes, but its DIME is much smaller compared to IMDb which has only two classes. This relative difference in difficulty of MNIST and IMDb is also reflected in their SOTA errors, validating our measure.

Finally, for datasets such as MNIST, Fashion MNIST, EMNIST, FakeData, and DBPedia, DIME provides a tight error bound. But for certain datasets such as CIFAR-10, CIFAR-100, and Tiny ImageNet, DIME seem slightly pessimistic. As we discussed in Remark 2, DIME could be larger than the true lower bound of $\mathcal{P}_e$ since we constrain the function space to be a neural network model. While the relative order of difficulty suggested by DIME is reasonable, its large value also suggests that an MLP may not have enough capacity for accurate $\mathcal{P}_e$ estimation. While it is an open question how to choose a model that is large enough to approximate $\mathcal{P}_e$ yet easy to optimize, Equation (5) suggests that as the model size grows, the gap between DIME and true lower bound of $\mathcal{P}_e$ becomes smaller. We verify this by calculating DIME with increasing number of neurons for a three-layer MLP. See Figure 4 in Appendix A.5 for details.

As we mentioned in the abstract, we can also use DIME to compute per-class difficulty score and reflect the effect of number of samples on data distribution, etc. See Appendix A.4 for the additional use cases for DIME.

## 4 Discussion

We would like to use this section to discuss some common questions proposed by the reviewers and readers.

1. Why don't we use a cross entropy estimation to approximate the conditional entropy $\mathbb{H}(Y|X)$ (which is also an upper bound of $\mathbb{H}(Y|X)$)?

   We want to emphasize that our motivation here is to design a metric for datasets to capture the intrinsic complexity of the sample-label distribution in supervised learning. Our intuition is that if a dataset has a large number of classes and the distributions of the features for each class are close to each other, then it would be difficult to accurately predict the labels given the features. Therefore, Our starting point is to define a metric of the following form:

   $$f(|\mathcal{Y}|) - \frac{1}{|\mathcal{Y}|} \sum_{y=1}^{|\mathcal{Y}|} D(X|Y=y||X) \tag{6}$$

   for some increasing function $f$ and some distance function $D$ over distributions. We find a good choice of $(f, D)$ is $(\log, \mathbb{KL})$ and this exactly leads to the definition of $\mathbb{H}(Y|X)$. Then we start to consider how to estimate the KL divergence for some high-dimensional data. Therefore, even if we can use a neural network via a cross entropy approach to approximate $\mathbb{H}(Y|X)$——and we expect the approximation would be more accurate than our approach as the distribution of $X|Y$ is generally much more complicated than that of $Y|X$——we still avoid this approach since it is just meaningless here, given that the spirit of this paper is to show that the distance over the distributions of features for each class is an essential indicator of the complexity of the label-feature distributions. Since our focus and interest are the distributions $X|Y=y$, an approach to approximate $Y|X$ is just another trial to do the prediction and thus, is not appropriate here to provide a model-free measure.

   On the other hand, we have to point out that one main limitation of our approach is that it is pretty challenging to approximate the distribution of $X|Y$ (i.e. achieving the supremum in (5)). A simple MLP neural network, as we used here, seems far from satisfying given that the fact that DIME is almost always larger than the SOTA error in Table 1. Therefore, it seems worthwhile to try some neural networks with more capacity or more complicated architectures like convnets, LSTMs, transformers, etc. (Of course, then it would be more controversial to claim our metric "model-free".)

2. Why do you use MINE (Belghazi et al. (2018)) to estimate the mutual information (which itself is not a valid lower bound, as shown by Poole et al. (2019))?

   We actually use a MINE style estimator to estimate the KL divergence instead of the mutual information. So some other methods to estimate the mutual information (see e.g. Alemi et al. (2016) for an upper bound; Oord et al. (2018) for a lower bound) are not exactly relevant here. That being said, it is definitely worthwhile to try these methods to directly estimate the mutual information (epsecially those for upper bounds such that we end up with a valid lower bound of $\mathbb{H}(Y|X)$) and compare the results with the current ones. We leave this for future explorations.

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

# A Supplementary Results for Experiments

## A.1 Algorithm for DIME

---

**Algorithm 1** DIME

---

Initialize $|\mathcal{Y}|$ neural networks: $T_{\theta_1}, T_{\theta_2}, \ldots, T_{\theta_{|\mathcal{Y}|}}$, one per class.
Initialize class counters $k_1 = 0, \ldots, k_{|\mathcal{Y}|} = 0$, one per class. Initialize the sample counter $k = 0$.
**for** $t$ in $0, \ldots, T$ **do**
    Draw a batch of $b$ training examples $\{(x_i, y_i)\}$, and $b$ evaluation examples $\{(x_i^e, y_i^e)\}$.
    **for** each class $c \in \{1, \ldots, |\mathcal{Y}|\}$ **do**
        Find samples of the $c$-th class: $S_c = \{i | y_i = c\}$, $S_c^e = \{i | y_i^e = c\}$
        Train $\theta_c = \arg\max_{\theta_c} \frac{1}{|S_c|} \sum_{S_c} T_{\theta_c}(x_i) - \log[\frac{1}{b} \sum_j \exp T_{\theta_c}(x_j)]$
        Evaluate $\widehat{\mathbb{KL}}_{\theta_c} = \frac{1}{|S_c^e|} \sum_{S_c^e} T_{\theta_c}(x_i^e) - \log[\frac{1}{b} \sum_j \exp T_{\theta_c}(x_j^e)]$
        If $\widehat{\mathbb{KL}}_{\theta_c}$ stops increasing, pause the training for $T_{\theta_c}$.
        Update counters: $k_c + = |S_c^e|$.
    **end for**
    Update the sample counter: $k + = b$
    If all neural networks stop updating, break.
**end for**
Estimate the class probability $\widehat{p}_c = \frac{k_c}{k}$, $c = \{1, \ldots, |\mathcal{Y}|\}$.
Calculate $\widehat{\mathbb{H}}(Y|X)$ using (4), $\widehat{p}_c$, and $\widehat{\mathbb{KL}}_{\theta_c}$, $c = \{1, \ldots, |\mathcal{Y}|\}$.
Solve (3) and output the solution.

---

## A.2 Supplement Results for Section 3.1

**Experiments with toy datasets:** We generate three two-dimensional 2-class toy datasets with 20,000 samples each with increasing level of difficulty (Figure 1). Samples in Toy Data 1 and Toy Data 2 are generated using random Gaussian variables by scaling differently in different dimensions and then by rotating, while the two classes in Toy Data 3 are simply uniform and Gaussian random variables. Toy Data 1, which is relatively easy to be separated, gets a DIME of 0.001. Toy Data 2, which is a bit harder due to a small overlapping region around the axis origin, gets a DIME of 0.013. And Toy Data 3, which is the hardest due to overlapping supports of the two class, gets a DIME of 0.263. In sum, DIME accurately reflects the relative difficulty of the toy examples.

**Label corruption test:** Next, we estimate DIME for the case of label corruption. If we assign random labels to a fraction of the training samples, intuitively the dataset should become more difficult. We experiment with five datasets with ten classes each—MNIST, Fashion MNIST, CIFAR-10, EMNIST (digits), and SVHN—and perform label corruption on their training sets in steps of 0.1 from 0 to 1. We observe a consistent increase in DIME across all the datasets with increasing label corruption (Figure 2-(left)). For some datasets such as MNIST, DIME is almost tight given the fraction of randomly corrupted labels.

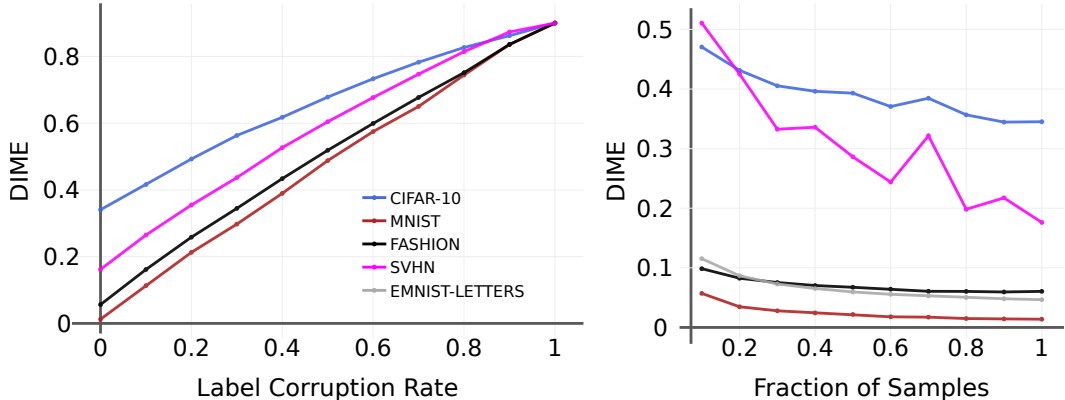

Figure 2: DIME increases with increase in the amount of label corruption (left). DIME becomes reasonably stable as the fraction of samples from the dataset used for estimation increase (right).

## A.3 Supplement Results for Section 3.2

### A.3.1 Choice of Benchmark Datasets

We evaluate the following image classification datasets: MNIST (LeCun et al., 1998), Extended MNIST (EMNIST) (Cohen et al., 2017), Fashion MNIST (Xiao et al., 2017), CIFAR-10 (Krizhevsky, 2009), CIFAR-100 (Krizhevsky, 2009), Tiny ImageNet[3], and SVHN (Netzer et al., 2011). The EMNIST dataset has several different splits, which include splits by digits, letters, merge, class, and balanced. For a controlled evaluation study, we also add the PyTorch Fakedata dataset[4] which contains 10-classes of randomly-generated $32 \times 32$ grayscale images.

Among natural language processing datasets, we investigate three supervised scenarios: sentiment analysis, text classification, and language modeling. For sentiment analysis, we analyze IMDb (Maas et al., 2011), and Stanford Sentiment Treebank (SST) (Socher et al., 2013). In the case of SST, we utilize both fine-grained labels (SST-5) and binary labels (SST-2). For the text classification task, we use the TREC dataset (Li & Roth, 2002), AG News, DBPedia, as well as Yelp Review dataset (Zhang et al., 2015). For language modeling task, we analyze the character level modeling for Penn Tree Bank (Marcus et al., 1994).

For DIME estimation, we do not perform any preprocessing on the data except for simple rescaling. For image data we rescale all the pixel values to floats in $[0, 1]$. By data processing inequality (Cover & Thomas, 2012), the rescaling will not cause any change on $\mathbb{H}(Y|X)$ and $\mathcal{P}_e$ since the rescaling function is invertible.

### A.3.2 Choice of Neural Network Estimator

To estimate DIME for image datasets, we use a simple multi-layer perceptron (MLP) with residual layers with Rectified Linear Unit (ReLU) activation functions. The network has three hidden layers of 4096 neurons. For language datasets, we use an embedding layer of dimension 1500, followed by three layers of 256 hidden neurons with residual connections and ReLU activations. We found that a higher dimension of the embedding helps achieve a tighter bound. The embedding layer is initialized with a concatenation of five pretrained embedding vectors: GloVe-840B-300d, GloVe-42B-300d, GloVe-twitter-27B-200d, GloVe-6B-300d, FastText-en-300d, and CharNGram-100d. The input sequences are truncated or padded to the length of 128. The MLP is operating on the token dimension instead of the embedding dimension to take the order of the sequence into consideration. For example, the first layer of MLP is 128-by-256 instead of 1500-by-256. The last fully-connected layer flattens everything into a vector and projects it to a scalar. For optimizing the MLP, we use SGD with initial learning rate of 0.1 and anneal it to 0.01 and 0.001 if the objective stops updating. We use the test set as evaluation set, as described in Algorithm 1.

---

[3]https://tiny-imagenet.herokuapp.com/
[4]https://github.com/pytorch/vision/blob/master/torchvision/datasets/fakedata.py

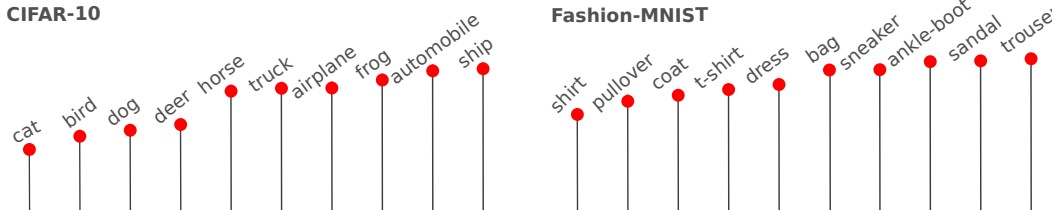

Figure 3: DIME can rank classes within datasets for difficulty. A lower height of the bar indicates higher relative difficulty.

### A.4 Additional Use Cases for DIME

**Dataset difficulty per class:** Algorithm 1 can also estimate the relative difficulty of classes within a dataset. For this purpose, we simply rank the classes by their $\widehat{\mathbb{KL}}(X|y\|X)$. A smaller value of $\widehat{\mathbb{KL}}(X|y\|X)$ indicates the class is harder to classify. As examples in Figure 3 show, 'cat' and 'bird' are the most difficult classes in CIFAR-10, while 'automobile' and 'ship' are the easiest. In addition, 'shirt' and 'pullover' are the most difficult classes in Fashion MNIST, while 'sandal' and 'trouser' are the easiest.

**Controlled Study with CIFAR Subsets:** We obtain a 10-class subset from CIFAR-100 by choosing subclasses from the 'large carnivores' and 'large omnivores and herbivores' superclasses. These classes—bear, leopard, lion, tiger, wolf, camel, cattle, chimpanzee, elephant, and kangaroo—should be intuitively harder to classify as compared to CIFAR-10, which includes disparate classes such as bird, truck, and ship. Note that the image data for both of these datasets comes from the same, larger '80 million tiny images' dataset (Torralba et al., 2008). We find that the DIME for this 10-class CIFAR-100 subset is 0.504, significantly larger than a DIME of 0.340 for CIFAR-10.

**Effect of number of samples:** With increasing number of samples, the empirical distribution of both training and testing tend towards the population distribution, making the distance between them smaller. It should also make the dataset easier, which is indeed reflected in a decreasing value of DIME with increasing number of samples (Figure 2-(right)). But we also observe that our measure becomes stable with increasing number of samples. This suggests that our method can be used to estimate the relative difficulty of new datasets in domains where data is hard to collect (e.g., medical datasets, fine-grained image classification) and make determinations if additional data collection is required to make the problem easier to solve.

### A.5 Model Complexity Experiment

Equation (5) suggests when the model class is larger the gap between DIME and the true lower bound on $\mathcal{P}_e$ becomes smaller. This is verified in our experiment shown in the middle of Figure 4, where we use neural networks with three hidden layers of varying number of neurons to calculate DIME. In the experiment we try 32, 64, 128, 256, 512, 1024, and 2048 neurons for each hidden layer. As the number of hidden neurons grows, DIME roughly decreases as expected. Still on CIFAR-10 the decrease is pretty monotonic. For simple datasets such as MNIST, Fashion MNIST and EMNIST (byletters) the numbers got stabilized pretty quickly.

## B Hybrid conditional entropy

Consider the usual classification setting, $X$, representing the feature, is a random vector in $\mathbb{R}^p$; $Y$, representing the label, is a random variable(or vector) in $\{1, \ldots, |\mathcal{Y}|\}$. Assume $X$ is a continuous variable with density $p_X(x)$ for $x$ in some compact domain $\Omega \subset \mathcal{X} \subset \mathbb{R}^p$. Can we define a hybrid conditional entropy $\mathbb{H}(Y|X)$ while $X$ is continuous and $Y$ is discrete? We give a formal definition for this mixed-pair case and demonstrate that $\mathbb{H}(Y|X)$ defined in our way can be a good indicator for the hardness of classification on the data $(X, Y)$, which is consistent with the classical definition in the sense that both of them give the intuition that how much information or uncertainty is left for $Y$ given $X$.

Our definition is

$$\mathbb{H}(Y|X) := \mathbb{E}_X \left[ -\sum_{y=1}^{|\mathcal{Y}|} \mathbb{P}(Y = y|X) \log \mathbb{P}(Y = y|X) \right], \qquad (7)$$

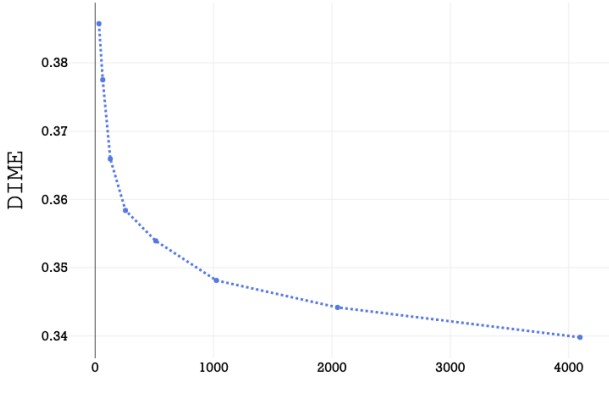

(a) DIME decreases on CIFAR-10 as the model size grows.

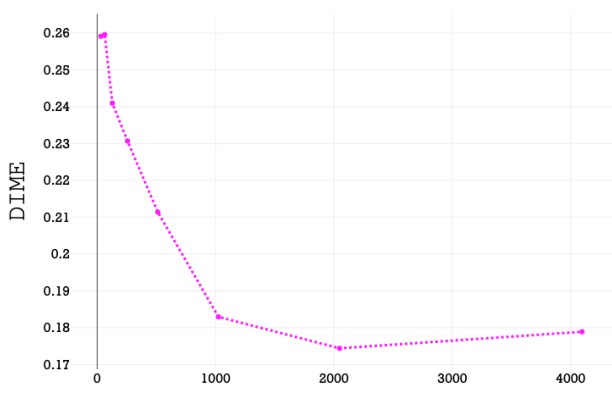

(b) DIME decreases on SVHN as the model size grows.

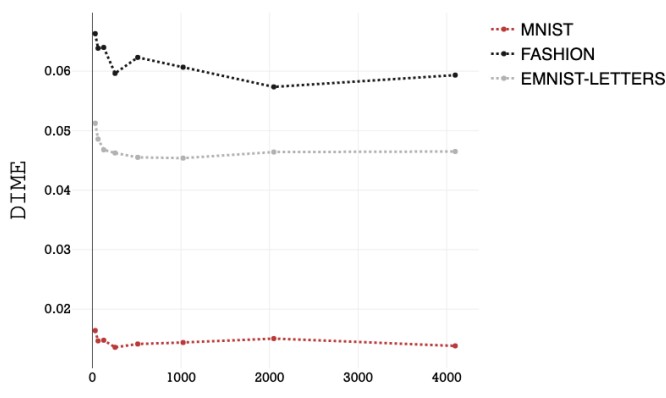

(c) DIME decreases on MNIST, Fashion MNIST, and EMNIST(byletters) as the model size grows.

Figure 4: DIME as the model size increases.

where $\mathbb{P}(Y = y|X) := \mathbb{E}[\mathbf{1}(Y = y)|X]$. By the definition of conditional expectation (or probability), it is easy to check that

$$\mathbb{E}[\mathbf{1}(Y = y)|X] = \frac{P(Y = y)p(X|Y = y)}{p_X(X)}, \tag{8}$$

where $p(x|Y = y), x \in \mathbb{R}^p$ is the conditional density function of $X$ given $Y = y$. Denote $p_y := P(Y = y)$, we have

$$\mathbb{H}(Y|X) = \int_\Omega -\sum_{y=1}^{|\mathcal{Y}|} p_y p(x|Y = y) \log \frac{p_y p(x|Y = y)}{p_X(x)} dx$$

$$= -\sum_{y=1}^{|\mathcal{Y}|} p_y \log p_y - \sum_{y=1}^{|\mathcal{Y}|} p_y \int_\Omega p(x|Y = y) \log \frac{p(x|Y = y)}{p_X(x)} dx$$

$$= \mathbb{H}(Y) - \sum_{y=1}^{|\mathcal{Y}|} p_y \mathbb{KL}(X|Y = y||X). \tag{9}$$

This proves (4).

We summerize some basic properties of $\mathbb{H}(Y|X)$ under our definition (7):

**Lemma 1.** *(i) $0 \leq \mathbb{H}(Y|X) \leq \mathbb{H}(Y) \leq \log|\mathcal{Y}|$.*

*(ii) $\mathbb{H}(Y|X) = \mathbb{H}(Y)$ if and only if $X$ and $Y$ are independent.*

*(iii) $\mathbb{H}(Y|X) = 0$ if and only if $Y$ is a function of $X$. That is, there exist $|\mathcal{Y}|$ subsets of $\Omega$: $\Omega_1, \ldots, \Omega_{|\mathcal{Y}|}$ such that $P(X \in \Omega_y \cap \Omega_{y'}) = 0$ for any $y \neq y' \in \{1, \ldots, |\mathcal{Y}|\}$, $P(X \in \cup_{y=1}^{|\mathcal{Y}|}\Omega_y) = 1$ and $P(Y = y|X \in \Omega_y) = 1$ for any $y \in \{1, \ldots, |\mathcal{Y}|\}$.*

*Proof.* (i) This is trivial by (7), (9) and the fact that KL divergence is always nonnegative.

(ii) By (9), it is easy to get

$$\mathbb{H}(Y|X) = \mathbb{H}(Y) \iff X|Y = y \overset{d}{=\!=} X, \quad \forall y \in \{1, \ldots, |\mathcal{Y}|\}.$$
$$\iff P(X \in A, Y = y) = P(X \in A)P(Y = y), \text{for any}$$
$$\text{measurable set } A \text{ in } \mathcal{X}, \text{any } y \in \{1, \ldots, |\mathcal{Y}|\}.$$
$$\iff X, Y \text{ are independent.}$$

(iii) **Sufficiency:** If $Y$ is a function of $X$, then by assumptions, for any $y_1 \neq y_2$, we have
$$P(X \in \Omega_{y_1}, Y = y_2) \leq P(X \in \Omega_{y_1}) - P(X \in \Omega_{y_1}, Y = y_1) = 0.$$

Therefore, $p(x|Y = y_2) = 0$ for $x \in \Omega_{y_1}$(a.e.). Then for any $y \in \{1, \ldots, |\mathcal{Y}|\}$, we have

$$\mathbb{KL}(X|Y = y) = \int_{\Omega_y} p(x|Y = y) \log \frac{p(x|Y = y)}{p_X(x)} dx \tag{10}$$

Notice that for $x \in \Omega_y$(a.e.), $p_X(x) = \sum_{y'=1}^{|\mathcal{Y}|} p_{y'} p(x|Y = y') = p_y p(x|Y = y)$, and thus

$$\mathbb{KL}(X|Y = y) = -\log p_y. \tag{11}$$

Then by (9), $\mathbb{H}(Y|X) = 0$.

**Necessity:** Assume $\mathbb{H}(Y|X) = 0$. Notice that $p_X(x) = \sum_{y=1}^{|\mathcal{Y}|} p_y p(x|Y = y), \forall x \in \Omega$. Therefore, for any $y \in \{1, \ldots, |\mathcal{Y}|\}$,

$$\mathbb{KL}(X|Y = y||X) = \int_\Omega p(x|Y = y) \log \frac{p(x|Y = y)}{p_X(x)} dx$$

$$\leq \int_\Omega p(x|Y = y) \log \frac{p(x|Y = y)}{p_y p(x|Y = y)} dx \tag{12}$$

$$= -\log p_y.$$

Combing (9) and (12), we know if $\mathbb{H}(Y|X) = 0$, then

$$p_X(x) = p_y p(x|Y = y), \forall x \in \{x \in \Omega : p(x|Y = y) > 0\} (a.e.), \forall 1 \leq y \leq |\mathcal{Y}|. \quad (13)$$

Define $\Omega_y := \{x \in \Omega : p(x|Y = y) > 0\}$, for every $y \in \{1, \ldots, |\mathcal{Y}|\}$. Without loss of generality, we assume $p_X(x) > 0$ for every $x \in \Omega$. (Otherwise, we can replace $\Omega$ by $\Omega \cap \{p_X(x) > 0\}$ and the following argument is still true.) Then we have

$$P(X \in \cup_{y=1}^{|\mathcal{Y}|} \Omega_y) = P(X \in \Omega) = 1.$$

By (13), we have

$$P(X \in \Omega_y) = \int_{\Omega_y} p_X(x)dx = \int_{\Omega_y} p_y p(x|Y = y)dx \quad (14)$$
$$= p_y P(X \in \Omega_y | Y = y) = P(X \in \Omega_y, Y = y).$$

Therefore, $P(Y = y | X \in \Omega_y) = 1$. Lastly, we show $P(X \in \Omega_y \cap \Omega_{y'}) = 0$ for any $y \neq y'$. This is immediate if we notice that by replacing $\Omega_y$ in (14) with $\Omega_y \cap \Omega_{y'}$, we can have

$$P(X \in \Omega_y \cap \Omega_{y'}) = P(X \in \Omega_y \cap \Omega_{y'}, Y = y).$$

Changing the postion of $y$ and $y'$, we have

$$P(X \in \Omega_y \cap \Omega_{y'}) = P(X \in \Omega_y \cap \Omega_{y'}, Y = y').$$

Thus, $P(X \in \Omega_y \cap \Omega_{y'}) = 0$.
Before the end of the proof, we want to add a remark that the condition $P(X \in \Omega_y \cap \Omega_{y'}) = 0$ is actually just a consequence of $P(Y = y | X \in \Omega_y) = 1, \forall y$. We add it explicitly in the conditions to make it clear that $\Omega_1, \ldots, \Omega_{|\mathcal{Y}|}$ is a partition of $\Omega$ according to the value of $Y$.

$\square$

## C  Proof of Fano's Inequality for Continuous Features

We see the above essential properties for $\mathbb{H}(Y|X)$ have been preserved for the mixture case. Furthermore, we show Fano's inequality still holds for our case. Our proof here is based on the proof of Lemma 2.10, Tsybakov (2004).

*Proof.* (Proof of Theorem 1)
Assume $\mathcal{P}_e(f) := P(f(X \neq Y))$. Then

$$\mathcal{P}_e(f) = \sum_{y=1}^{|\mathcal{Y}|} P(Y = y)P(f(X) \neq y|Y = y)$$

$$= \sum_{y=1}^{|\mathcal{Y}|} P(Y = y) \int I(f(x) \neq y)p(x|Y = y)dx$$

$$= \int \left[ \sum_{y=1}^{|\mathcal{Y}|} P(Y = y)I(f(x) \neq y)\frac{p(x|Y = y)}{p(x)} \right] p(x)dx,$$

where $p(x|Y = y)$ is the density function of $X|Y = y$ and $p(x)$ is the density of $X$. Denote

$$p(y|x) := \frac{P(Y = y)p(x|Y = y)}{p(x)},$$

we have

$$\mathcal{P}_e(f) = \int \left( \sum_{y \neq f(x)} p(y|x) \right) p(x)dx \quad (15)$$

Define $g(x) := \mathbb{H}(x) + x \log(|\mathcal{Y}| - 1)$, for $x \in [0, 1]$. Here, $\mathbb{H}(x) = -x \log x - (1 - x) \log(1 - x)$. As $g$ is a concave function, by Jensen's Inequality, we have

$$g(\mathcal{P}_e(f)) \geq \int g\left( \sum_{y \neq f(x)} p(y|x) \right) p(x)dx \tag{16}$$

We claim:

$$g\left( \sum_{y \neq f(x)} p(y|x) \right) \geq -\sum_{y=1}^{|\mathcal{Y}|} p(y|x) \log p(y|x), \quad \forall x. \tag{17}$$

If the claim (17) is true, then

$$g(\mathcal{P}_e(f)) \geq \int -\sum_{y=1}^{|\mathcal{Y}|} p(y|x) \log p(y|x) p(x)dx = \mathbb{H}(Y|X). \tag{18}$$

By the definition of $\mathcal{P}_e$, there exists a sequence $\{f_k\}_{k=1}^{\infty}$ such that $\mathcal{P}_e(f_k) \to \mathcal{P}_e$. Therefore,

$$g(\mathcal{P}_e) = \lim_{k \to \infty} g(\mathcal{P}_e(f_k)) \geq \mathbb{H}(Y|X). \tag{19}$$

This is exactly (3). So it suffices to show the claim (17) is true. We may assume that $\sum_{y \neq f(x)} p(y|x) \neq 0$ (otherwise, it is trivial). Let $y_0 = f(x)$, then

$$\sum_{y=1}^{|\mathcal{Y}|} p(y|x) \log p(y|x) = p(y_0|x) \log p(y_0|x) + \left( \sum_{y \neq y_0} p(y|x) \right) \log \left( \sum_{y \neq y_0} p(y|x) \right)$$

$$+ \sum_{y \neq y_0} p(y|x) \log \frac{p(y|x)}{\sum_{y' \neq y_0} p(y'|x)}$$

$$= -\mathbb{H}\left( \sum_{y \neq y_0} p(y|x) \right) + \left( \sum_{y \neq y_0} p(y|x) \right) \sum_{y \neq y_0} q(y|x) \log q(y|x),$$

where

$$q(y|x) := \frac{p(y|x)}{\sum_{y' \neq y_0} p(y'|x)}, \quad \sum_{y \neq y_0} q(y|x) = 1, \quad q(y|x) \geq 0. \tag{20}$$

Then by Jensen's Inequality, we have

$$\sum_{y \neq y_0} q(y|x) \log q(y|x) \leq \log(|\mathcal{Y}| - 1). \tag{21}$$

Therefore,

$$\sum_{y=1}^{|\mathcal{Y}|} p(y|x) \log p(y|x) \leq -\mathbb{H}\left( \sum_{y \neq y_0} p(y|x) \right) + \left( \sum_{y \neq y_0} p(y|x) \right) \log(|\mathcal{Y}| - 1) \tag{22}$$

$$= -g\left( \sum_{y \neq y_0} p(y|x) \right). \tag{23}$$

$\square$

