# OpenReview forum: "DIME: An Information-Theoretic Difficulty Measure for AI Datasets"
_NeurIPS.cc/2020/Workshop/DL-IG — NeurIPSW 2020: DL-IG Poster_

### Official Review · AnonReviewer2 · 2020-10-26
**Review of "DIME: An Information-Theoretic Difficulty Measure for AI Datasets"**

**Rating:** 5
**Confidence:** 5

**Review:**

This paper considers an approach to measure the "difficulty" of modeling a dataset. I think the motivation and approach are ill-conceived. However, there may be some value in exploring the proposed measure for other purposes.

- Sample complexity and the complexity of the hypothesis space ("distributional complexity") are quite related, this is the heart of statistical complexity theory and ideas like VC dimension.

- The most glaring issue is that in Eq. 3, the most natural estimator for H(y|x) is to lower bound it with the cross entropy. But this reveals the tautological nature of this paper: a difficult problem is one where it is difficult to accurately predict the labels.

- The fact that FakeData is the hardest confirms the problem with the set-up. If you were to try to predict completely random labels, your prediction error is maximal and therefore this task is the "most difficult". Yet, in some ways this problem is not difficult at all, as any learner will have the same performance on this task!

- The MINE estimator has a number of issues, other variations are better. See Poole et al. "On variational bounds of mutual information"

- I don't believe we can learn much from Table 1, except the extent to which your D-V approximation leads to a sub-optimal classifier compared to SOTA ones. (Note that T in Eq. 5 can be interpreted as a classifier - so you are essentially just doing a prediction with a different form for the loss function.)

At the end, you mention an idea which might be worth pursuing with the approach: characterizing a per-class difficulty score. I still don't think "difficulty" is the right word, but it would at least tell you if there are significant differences between the shapes of the distributions for each class. You could interpret this as measuring the classes that have the most variation in a dataset. In medical domains, for instance, this could be useful, as it might signal that one disease has a lot more variation, and potentially consists of multiple distinct disorders that have not yet been recognized.

---

### Official Review · AnonReviewer2 · 2020-10-31
**A proposed metric for dataset hardness.**

**Rating:** 3
**Confidence:** 5

**Review:**

While I applaud the goal of the paper, I have some issues with the currently formulation of the paper.

It suggests a measure of dataset difficulty which amounts to building an estimate of the mutual information in order to use Fano's inequality to estimate a lower bound on the optimal error rate achieved.  Unfortunately the bounds are broken.  Fano's inequality gives us a lower bound on the minimum error probability in terms of the conditional probability.  We would therefore need to be able to lower bound the conditional entropy to provide a valid bound.  Since the conditional entropy is inversely proportional to the mutual information we would therefore need an upper bound on the mutual information.  The paper then uses a MINE style estimator the mutual information which itself purports to be a *lower* bound.  This bound goes the wrong way in order to accomplish its goal. In addition the style of estimator used here (MINE) has been demonstrated to not actually be a valid bound itself, further complicating the story. (see. e.g. Poole et al. "On variational bounds of mutual information")

If what we wanted was an upper bound on the conditional entropy, we could have just looked at the conditional likelihood of any trained model, as any trained models likelihood provides a valid upper bound on the conditional entropy.  As their table demonstrates, their estimator, which is supposed to provide a *lower* bound on the minimum error rate is actually higher than the reported results on all but one of the datasets, which itself has not really attracted much attention.  This just goes to show that their estimator is quite bad, and much worse at modelling the distributions than other trained models.  I understand the desire to have some model independent notion of hardness for each dataset, but it's not as if the MINE style estimator used here is independent of any modelling assumptions.  The fact that the bounds come out so poor is almost surely because a weak model was used to try to estimate what is a hard structured generation task.  I would be curious to see what sort of bounds are generated if you use a similarly weak mlp but instead use it to bound the conditional entropy directly with the cross entropy.  I suspect it actually works much better than the approach here.

Backing up for a second, even if we had good estimates of the conditional entropy, and therefore assumed we had good estimates of the mutual information, it's a bit unfair to call that a difficulty for the dataset.  By this metric a if the input and labels were independent this would be called 'difficult'.  It's surely difficult in the sense that the best anyone could ever do at prediction is only ever as good as random guessing, but this doesn't capture the semantic notion of difficulty that I think people want when they discuss datasets.

Minimally, I think the paper needs a substantial rewrite to point out some of these flaws with the method, lest the reader leaves with some misconceptions.

---

### Decision · Program_Chairs · 2020-11-07

**Decision:**

Accept (Poster)

**Comment:**

The authors should update the manuscript to incorporate the reviewer feedback, in particular a thorough rewrite of this submission and correct the estimate of the mutual information.